# Helical versus Flat Bis-Ferrocenyl End-Capped Peptides: The Influence of the Molecular Skeleton on Redox Properties

**DOI:** 10.3390/molecules27186128

**Published:** 2022-09-19

**Authors:** Saverio Santi, Barbara Biondi, Roberta Cardena, Annalisa Bisello, Renato Schiesari, Silvia Tomelleri, Marco Crisma, Fernando Formaggio

**Affiliations:** 1Department of Chemical Sciences, University of Padova, Via Marzolo 1, 35131 Padova, Italy; 2Institute of Biomolecular Chemistry, Padova Unit, CNR, Via Marzolo 1, 35131 Padova, Italy

**Keywords:** helical peptides, ferrocene, α-amino isobutyric acid, 2,3-diamino propionic acid, C^α,β^-didehydroalanine, macrodipole, electrochemistry, exciton-coupled Circular Dichroism

## Abstract

Despite the fact that peptide conjugates with a pendant ferrocenyl (Fc) have been widely investigated, bis-ferrocenyl end-capped peptides are rarely synthetized. In this paper, in addition to the full characterization of the Fc-CO-[_L_-Dap(Boc)]_n_-NH-Fc series, we report a comparison of the three series of bis-ferrocenyl homopeptides synthesized to date, to gain insights into the influence of α-amino isobutyric (Aib), 2,3-diamino propionic (Dap) and C^α,β^-didehydroalanine (ΔAla) amino acids on the peptide secondary structure and on the ferrocene redox properties. The results obtained by 2D NMR analysis and X-ray crystal structures, and further supported by electrochemical data, evidence different behaviors depending on the nature of the amino acid; that is, the formation of 3_10_-helices or fully extended (2.0_5_-helix) structures. In these foldamers, the orientation of the carbonyl groups in the peptide helix yields a macrodipole with the positive pole on the N-terminal amino acid and the negative pole on the C-terminal amino acid, so that oxidation of the Fc moieties takes place more or less easily depending on the orientation of the macrodipole moment as the peptide chain grows. Conversely, the fully extended conformation adopted by ΔAla flat peptides neither generates a macrodipole nor affects Fc oxidation. The utilization as electrochemical and optical (Circular Dichroism) probes of the two terminal Fc groups, bound to the same peptide chain, makes it possible to study the end-to-end effects of the positive charges produced by single and double oxidations, and to evidence the presence “exciton-coupled” CD among the two intramolecularly interacting Fc groups of the _L_-Dap(Boc) series.

## 1. Introduction

Bio-organometallic pendant ferrocenyl (Fc) peptide conjugates have been explored extensively [1,2,3,4,5,6,7,8,9,10,11,12,13,14,15,16,17,18,19,20,21,22]. This class of peptides has allowed researchers to design systems with new redox triggered functions, such as chirality organization [1,11], chirality dependence charge transfer rate [2], photodynamic anticancer [3,7], antibacterial [4,10], antimicrobial [6] and antimalarial [7] activities, supramolecular gelation [8,18], potential bioinspired electronic materials [9], changes in the self-assembly [12], sensory applications [13,14,19], enzymatic fuel cells [15,16] and bioelectrocatalysts [22]. In particular, we have employed a single Fc group as a scaffold to support peptide secondary structural motifs, such as 3_10_-helices with α-amino isobutyric acid (Aib) [23] or _L_-2,3-diamino propionic (_L_-Dap) acid [24], and the 2.0_5_-helix based on C^α,β^-didehydroalanine (ΔAla) [25]. 

Curiously, bis-ferrocenyl end-capped peptides have been seldom synthetized. To our knowledge, the unique two examples of homopeptides were reported by us, namely Fc–CO–(Aib)_n_–NH–Fc (*n* = 1–5) [26,27] and Fc–CO–(ΔAla)_n_–NH–Fc (*n* = 1–4) [28]. In addition, two dinuclear ferrocene-based conjugates comprising a sequence of two amino acids (Fc–CO–L-Pro–L-Ala–NH–Fc and Fc–CO–D-Pro–D-Ala–NH–Fc) were prepared and characterized by Nuskol et al. [29]. An intramolecular Fc–N–H···O=C–Fc hydrogen bond (β-turn) is responsible for dipeptide folding. 

The two terminal redox-active Fc groups allowed us to study the end-to-end effects of the positive charges produced by single and double oxidations using electrochemical techniques, the results of these studies indicating that charge transfer across the peptide main chain is deeply influenced by the nature and length of the peptide secondary structure. Helical peptides play a crucial role in mediating electron/charge transfer in biological systems [30,31,32,33,34,35,36,37,38,39,40]. The orientation of the carbonyl groups in a peptide helix yields a macrodipole with the positive pole on the N-terminal amino acid and the negative pole on the C-terminal amino acid. Therefore, the electron transfer is much faster from the C-terminal to the N-terminal site of a helix than in the opposite direction [41,42,43,44,45,46,47].

As precursors of the Fc–CO–(ΔAla)_n_–NH–Fc series, we first synthesized the corresponding _L_-Dap(Boc) oligomers [28]. Although Dap is a non-coded amino acid, it is a natural amino acid [48] observed in vegetal species and has also been discovered in meteorites [49]. Dap found application in the synthesis of new foldamers [50], nucleobase-conjugated peptides [51], as an organogelator [52], as an EPR probe for protein studies [53] and as a cell-penetrating peptide [54]. Moreover, Dap was reported to be a helix breaker [14], or a residue with complex conformational inclinations [55,56]. However, for the first time, we performed an in-depth analysis of the conformational preferences of Dap. Surprisingly, we found that the side-chain-protected Dap can fold into a β-turn or an incipient 3_10_-helix, as evidenced by our 2D NMR and X-ray crystal analyses, further supported by the electrochemical data [24].

In this paper, continuing our investigation of bis-functionalized Fc–peptide–Fc series, we have fully characterized the series of Fc–CO–(_L_-Dap)_n_–NH–Fc peptides, and discussed the close connection between structural properties and response to electrochemical stimuli. In addition, we have compared these outcomes with those previously obtained for the (Aib)_n_ and (ΔAla)_n_ series.

## 2. Results and Discussion

### 2.1. Peptide Synthesis

The synthesis of Fc–CO–[_L_-Dap(Boc)]_n_–NH–Fc (**1**–**4**) peptides (*n* = 1–4), as precursors for the preparation of the ΔAla series, was previously reported (Figure 1) [28]. 

Briefly, Z–_L_-Dap(Boc)–OH (Z, benzyloxycarbonyl; Boc, *t*-butyloxycarbonyl) was coupled to ferrocenyl amine by means of N-ethyl-N′-(3-dimethylamino-propyl)carbodiimide (EDC) and 1-hydroxybenzotriazole (HOBt) to obtain Z–_L_-Dap(Boc)–NH–Fc. After removal of the Z protecting group by catalytic hydrogenolysis, further Z–_L_-Dap(Boc)–OH derivatives were introduced with the same two steps (hydrogenolysis and EDC/HOBt coupling) procedure, obtaining Z–[_L_-Dap(Boc)]–NH–Fc, Z–[_L_-Dap(Boc)]_2_–NH–Fc, Z–[_L_-Dap(Boc)]_3_–NH–Fc and Z–[_L_-Dap(Boc)]_4_–NH–Fc. Finally, after removal of the Z protecting group, the N-terminal Fc–COOH moiety was introduced in all four peptides, obtaining Fc–CO–[_L_-Dap(Boc)]–NH–Fc (**1**) (yield 70%), Fc–CO–[_L_-Dap(Boc)]_2_–NH–Fc (**2**) (73%), Fc–CO–[_L_-Dap(Boc)]_3_–NH–Fc (**3**) (51%) and Fc–CO–[_L_-Dap(Boc)]_4_–NH–Fc (**4**) (50%) (Figure 1).

### 2.2. Crystal State Conformational Analysis

The molecular structure of **2**, as determined by single crystal X-ray diffraction, is illustrated in Figure 2. Relevant crystal data and structure refinement parameters are listed in Appendix A. Relevant backbone and side-chain torsion angles are reported in Appendix A. Intra- and intermolecular hydrogen bond parameters are listed in Appendix A.

In each of the Fc units, the two cyclopentadienyl rings are coplanar to each other (the angle between normals to their average planes is within 1.3(2)°), and they are found in a nearly eclipsed disposition, the values of the average inter-ring C=O···H–N twist angles being 8.9° and 9.2° for the N- and the C-terminal Fc unit, respectively. Such geometrical features are quite common for monosubstituted Fc derivatives [57,58]. The angle between normals to the cyclopentadienyl rings of the N- and C-terminal Fc units directly connected to the peptide backbone is 26.8(2)°. The distal cyclopentadienyl rings point in opposite directions relative to the peptide backbone. As a result, the Fe1 … Fe2 intramolecular separation, 8.99 Å, is much larger than that between the C01 and CT1 atoms (5.98 Å).

The peptide backbone is folded in a β-turn conformation [59] stabilized by an intramolecular C=O···H–N hydrogen bond between the N-terminal ferrocenoyl O0 carbonyl oxygen and the C-terminal amide NT-HT group. The N…O and H…O separations, (3.304(5) Å and 2.47 Å, respectively, (Appendix A)) are close to the upper limit commonly accepted for the occurrence of a C=O···H–N hydrogen bond, although the N–H…O angle, 163.3°, is normal [60,61,62]. Such a weak hydrogen bond might result, at least in part, from the participation of the O0 carbonyl oxygen to an intermolecular C–H…O interaction [63] with the co-crystallized CDCl_3_ molecule (Appendix A). Two additional intramolecular C=O···H–N hydrogen bond are observed. Specifically, for each of the _L_-Dap residues, the N^α^-H group is hydrogen bonded to the urethane carbonyl oxygen of the Boc moiety linked to the N^γ^ atom (Appendix A). Such backbone-to-side-chain hydrogen bonds close seven-membered pseudocycles.

All of the amide, peptide and urethane bonds are in the trans disposition, the largest deviation from planarity (|Δω| = 12.2°) being observed for the side-chain urethane bond of _L_-Dap(1) (Appendix A). The sets of φ, ψ backbone torsion angles adopted by Dap(1) [φ_1_, ψ_1_ = −79.4(5)°, −2.8(6)°] and _L_-Dap(2) [φ_2_, ψ_2_ = −79.9(5)°, −10.1(5)°] are not far from those typical for the *i* + 1 and *i* + 2 corner positions of a type-I β-turn (φ, ψ = −60°,−30° and −90°, 0°, respectively) [58]. 

### 2.3. Solution Conformational Analysis

We carried out a conformational analysis of **1**–**4** in CDCl_3_ solution by means of NMR and IR absorption spectroscopies.

Through 2D NMR spectra, we were able to completely assign the ^1^H resonances for all Fc–peptide conjugates (Appendix A). In particular, an in-depth analysis of the NOESY (spectrum of **2** (Figure 1)) in CDCl_3_ solution evidenced the intraresidue correlations between the αN–H and the side-chain N-H for both Dap(1) and Dap(2) (highlighted in magenta in Figure 3), strongly supporting the view that the intraresidue H-bonds found in the crystal state also persist in solution. 

Moreover, a cross peak between the ^α^C–H of Dap(1) and the N–H of the ferrocenyl unit at the *C*-terminus, and a correlation involving a proton of the *N*-terminus Fc (Fc1) and the N–H of Dap(2), suggests that Dap(1) adopts a folded/helical conformation. Additionally, in the amide region the sequential cross peak NH*_i_*→NH*_i+_*_1_ between Dap(1) and Dap(2) is observed, consistent with an incipient 3_10_-helix conformation.

Interestingly, the same conclusion is supported by the analysis of the NH chemical shifts behavior when a small amount of DMSO is added to the CDCl_3_ solution (Appendix A). Indeed, only the two β-NH on the side chain of the Dap residues are affected by DMSO, a strong H-bond acceptor. The other three NHs are insensitive to this addition, implying that they are already involved in H-bonds.

Homopeptide **3** (Figure 2) evidenced a similar pattern of connectivity, even if only the intraresidue correlation between the ^α^N–H and the side-chain N–H for Dap(1) is present. 

The fingerprint region of its NOESY spectrum shows long range cross peaks involving the ^α^C–H of Dap(2) and the N–H of the ferrocenyl unit (Fc2) at *C*-terminus, and a proton of the N-terminal Fc unit (Fc1) and the N–H of Dap(2) (Figure 4). Additionally, in the amide region the sequential cross peaks NH*_i_*→NH*_i+_*_1_ involving Dap(2) N–H, Dap(3) N–H and the N–H of the Fc unit at *C*-terminus can be identified. All these correlations confirm the adoption of a 3_10_-helical conformation. 

Unfortunately, the poor solubility of **4** in CDCl_3_ did not allow us to record 2D NMR spectra of good quality. Therefore, we opted for a DMSO-*d*_6_ solution where, unfortunately, the ability of DMSO to act as an H-bond acceptor appear to destabilize the helical structure of the peptide.

The conclusions of the NMR study are supported by the FT–IR analysis of CDCl_3_ solution (Figure 5). In the amide A region, all members of the **1**–**4** peptide series display a band at about 3450 cm^−1^, typical of NH groups not involved in H-bonding (Figure 5a), and indeed, related to the side-chain NH of the Dap residues, which do not form H-bonds.

Its intensity increases along the series because the number of Dap residues increases. The bands in the 3350–3330 cm^−1^ interval are assigned to NHs involved in H-bonds. Since the spectra show no significant changes upon dilution from 0.5 to 0.1 mM (Figure 5b), these bands are indicative of NH groups involved in intramolecular H-bonds. The absorptions centered at about 3300 cm^−1^ can be assigned to very strong H-bonds, while those typical of turn/helices are located in the 3350–3320 cm^−1^ range. Interestingly, the 3300 cm^−1^ band matches perfectly with the features of the X-ray crystal structure (Figure 2), where the intraresidue N–H···O=C H-bonds of the Dap residues appear to be stronger (H···O distances 2.11 and 2.07 Å) as compared to the H-bond closing the β-turn (H···O distance 2.47 Å) (Appendix A). 

The growing intensity of the band at about 3350 cm^−1^ in the longer peptides is attributed to the increasing number of β-turns formed. Therefore, these data are in agreement with the results obtained from the X-ray crystal and NMR analyses, and confirm that _L_-Dap(Boc) peptides adopt turn/helical 3D structures, stabilized by intramolecular H-bonds.

### 2.4. Cyclic Voltammetry Analysis

The characteristics of Fc, such as excellent stability in solution and air, flexibility in organic chemistry and a reversible electrochemical behavior, make it advantageous in labeling biomolecules [64]. In addition, electrochemistry on Fc-terminal peptides provides information on the effect of folding and length of the peptide chain [23,24,25,26,27,28]. 

Cyclic voltammetry (CV) of the bis-Fc-decorated peptides **1**–**4**, recorded under argon in CH_2_Cl_2_/0.1 M *n*Bu_4_NPF_6_ at potential scan rates in the range 0.1–5 Vs^−1^, shows two reversible, single-electron oxidation waves of the Fc unit (Figure 6). 

The oxidation potential of the Fc-NH is in the range of *E*_1/2_ = 0.29–0.36 V, while that of the Fc–CO is in the range *E*_1/2_ = 0.62–0.70 V versus SCE, as predictable on the basis of the electron donor–acceptor capability of the N–H and C=O groups, respectively. The intensity of peak current (*i*_max_) decreases along the peptide series. In general, larger molecules possess lower mobility towards an electrode. In fact, in a cyclic voltammetry experiment, *i*_max_ ∝ *D*^1/2^, and *D* depends on the molecular size, according to the Stokes–Einstein (*D* ∝ 1/*r*) equation [28]. Longer molecules are expected to display lower *D* values on the basis of the fluid dynamic effect. However, the equivalent radius *r* should not linearly vary with peptide length if a helical conformation is formed, as the number of residues increases.

As the _L_-Dap(Boc) chain grows (Figure 7), the Fc–NH moiety is oxidized more easily, and the potential decreases, whereas the Fc–CO moiety becomes more difficult to oxidize, and the potential increases.

This behavior is due to the peptide macrodipole that is oriented in the same or opposite direction with respect to the position of the Fc involved in the redox event [65]. In Figure 7, the potential variations of mono-Fc peptides _L_-Dap(Boc) [24] are also reported. Similar slopes are calculated for the oxidation of Fc–NH in the **1**–**4** (−21 ± 4 mV) and Bz-[_L_-Dap (Boc)]_n_–NH-Fc series (−23 ± 2 mV), and of Fc–CO in **1**–**4** (27 ± 5 mV) and Fc–CO–[_L_-Dap(Boc)]_n_–NH–iPr (28 ± 3 mV).

As evidenced by the X-ray structure (Figure 2) and the NOESY spectra (Figure 3 and Figure 4), the backbone is folded in a β-turn conformation stabilized by (i) an end-to-end intramolecular C=O···H–N hydrogen bond between the N-terminal ferrocenoyl O0 carbonyl oxygen and the C-terminal ferrocenoyl amide NT-HT group, and (ii) by intraresidue H-bonds between the αN–H and the side-chain N–H for both _L_-Dap(Boc) residues, establishing a particularly efficient electron-donating shortcut.

The comparison of the potential variations with those of the unique bis-Fc homopeptide examples reported before, Fc–CO–(Aib)_n_–NH–Fc (*n* = 1–5) [25,26] and Fc–CO–(ΔAla)_n_–NH–Fc (*n* = 1–4) [28], allows us to shed some light on how the charge transfer across the peptide main chain is influenced by the nature of the peptide secondary structure, exploiting the effect of the positive charges produced by single and double oxidations (Figure 8).

Mainly, it appears that in the 3_10_-helical homopeptides of *L*-Dap(Boc) and Aib amino acids, characterized by active dipole moments, an important and continuous variation of *E*_1/2_ occurs as the peptide length is increased, positive and negative depending on the orientation of the peptide dipole moment with respect to Fc. On the contrary, ΔAla homopeptides, adopting the fully extended conformation (2.0_5_-helix) [66,67], have no dipole moment, and consequently, the terminal Fcs show negligible variations of their *E*_1/2_.

In addition, while the oxidation potential of the C-terminal Fc in the _L_-Dap(Boc) and Aib series decreases with the elongation of the peptide chain, displaying almost identical slopes (d*E*_1/2_/d*n* = 22 ± 4 and 23 ± 3 mV), the potential of the N-terminal Fc increases more rapidly for the _L_-Dap(Boc) (d*E*_1/2_/d*n* = 27 ± 6 mV) than that of the (Aib)_n_ homopeptides (d*E*_1/2_/d*n* = 8 ± 1 mV). This result is rather surprising as it seems to suggest that _L_-Dap(Boc) homopeptides form helices more stable than those of Aib homopeptides. Indeed, Aib is known to be one of the strongest helix inducers. 

Therefore, we believe that the higher efficiency of _L_-Dap(Boc) in carrying the positive charge along the peptide chain is due to two contributions: the dipole moment generated by the helical arrangement and the peculiar, intraresidue H-bond (main-to-side chain) described above. 

### 2.5. Vis–MIR Chemical Oxidation

The formation of the oxidized species was detected in the visible (Figure 9) and IR (Figure 10) regions.

Stable solutions of **1^+^**–**4^+^** and **1^2+^**–**4^2+^** in CH_2_Cl_2_/0.1 M *n*Bu_4_NPF_6_ were obtained by the successive addition of increasing amounts of ferrocenium[BF_4_] and acetylferrocenium[BF_4_] (up to 1 equivalent for each oxidizing agent) to the neutral peptides. In the visible region, an absorption band appears around 758 nm after the addition of 1 equivalent of ferrocenium[BF_4_], characteristic of the Fc^+^–NH group. After the successive addition of 1 equivalent of acetylferrocenium[BF_4_], the typical absorption band around 620 nm of the Fc^+^–CO group is observed [24], while the intensity of the low-energy band decreases. For example, the visible spectra obtained by the stepwise oxidation of **4** (Figure 9, bold black line) clearly show the initial formation of the **4^+^** monocation (bold green line), which subsequently evolves into the **4^2+^** dication (bold red line).

The same solutions employed for the visible analysis of **1^+^**–**4^+^** and **1^2+^**–**4^2+^** produced the IR spectra reported in Figure 10. In both amide A (left) and carbonyl (right) regions, the spectra relative to the mono- and dications were rather similar and almost superposable to those found and already discussed for the mono-Fc series Bz–[_L_-Dap(Boc)]_n_–NH–Fc^+^ (*n* = 1–4) [24]. 

In a nutshell, the presence of the N–H bands around 3300 cm^−1^ (Figure 10a) points out that the intramolecular network of C=O···H–N bonds is affected by the oxidation, but largely preserved. In particular, in the **1^+^**–**4^+^** and **1^2+^**–**4^2+^** spectra (Figure 10b,c), the N–H absorptions at about 3300 cm^−1^, ascribable to the main-to-side chain H-bonds, are preserved. 

Conversely, the absorptions at about 3350 cm^−1^, due to β-turn-forming H-bonds, move to higher energies but do not disappear. Considering that these N–Hs are aligned with the dipole moment of the helix, their stretching is more sensitive to the positive charge introduced. These findings suggest that the second charge does not affect the efficiency of the _L_-Dap(Boc)-based 3_10_-helices in transmitting the positive charge along the peptides through the H-bond network, even in the longest **4**. 

This is in agreement with the CV results (Figure 7), as the potential values and their variations along the series **1^2+^**–**4^2+^** and Bz–[_L_-Dap(Boc)]_n_–NH–Fc were almost identical.

### 2.6. Circular Dichroism Analysis

A chiral induction is promoted on the achiral Fc by the covalently bound peptide. We therefore carried out a Circular Dichroism (CD) analysis in CH_2_Cl_2_, the same solvent used for the oxidation and electrochemical measurements. Both the solvent and Fc absorb in the amide region (190–250 nm), thus hampering analysis in the typical peptide zone. However, interesting information could be obtained in the 300–600 nm region. The absorptions of Fc in this interval are presented in Appendix A, whereas Figure 11a reports the CD spectra of compounds **1**–**4** in this region.

To better understand the behavior observed, we also built 3_10_-helical models for our compounds (Appendix A). The intramolecular Fc–Fc distances in **1**–**4** are, respectively, 5.06, 5.33, 7.48 and 9.70 nm. One would expect similar spectra for the four compounds, all having one Fc–CO- and one NH–Fc unit. Conversely, the spectra of **2** and **3** are almost overlapping, but those of **1** and **4** diverge significantly. 

This behavior calls for an Fc–Fc interaction. Curiously, **4** displays the largest intensity, despite having the Fc couple with the greatest separation (9.7 nm). 

We are quite confident in assigning this outcome to the “exciton-coupled” CD among the two Fcs intramolecularly interacting. Two pieces of evidence support such a hypothesis: (i) the CD signal almost disappears when one of the two Fcs is removed (Figure 11b); (ii) in **2** and **3,** a characteristic, bisignate curve is observed, with a positive maximum at 417 nm, a negative maximum at 468, and a crossover point corresponding to the absorption maximum of that transition (about 440 nm). This split signal is less evident in **4**, but a very strong interaction is observed in this case as well. Based on previous experimental data (helical peptides conjugated to two porphyrins) [68], we conclude that these interactions depend not only on the Fc–Fc distance, but also on the reciprocal orientation of the two aromatic units (parallel or perpendicular) and, of paramount importance, on the peptide H-bond scheme. Indeed, this latter can offer through-bond shortcuts to go from one Fc to the other Fc, as in the case of **2** and **4** (Appendix A).

## 3. Materials and Methods

### 3.1. Nuclear Magnetic Resonance

The ^1^H spectra were obtained on a Bruker Avance III HD spectrometer operating at 400.13 (***T*** = 25 °C). The peptide concentration in solution was 1 mM in spectrograde CHCl_3_-d_1_ (99.8% d_1_ containing 0.5 wt. % of silver foil as stabilizer and 0.03% (*v/v*) tetramethylsilane; Sigma-Aldrich, Milano, Italia) and DMSO-d_6_ (99.96%; D-Eurisotop). Processing and evaluation of the experimental data was carried out using the TOPSPIN software packages. All homonuclear spectra were acquired by collecting 400 experiments, each consisting of 32 scans and 2K data points. The spin systems of the amino acid residues were identified using standard chemical shift correlation and 2D (NOESY, TOCSY and COSY) experiments. 

### 3.2. X-ray Diffraction

Crystals of **2** were grown by slow evaporation from a CDCl_3_ solution in an NMR tube. X-ray diffraction data were collected with a Gemini E four-circle kappa diffractometer (Agilent Technologies) equipped with a 92 mm EOS CCD detector, using graphite-monochromated Mo Kα radiation (λ = 0.71073 Å). Data collection and reduction was performed with the CrysAlisPro software system (Rigaku Oxford Diffraction). A semi-empirical absorption correction based on the multi-scan technique using spherical harmonics, implemented in the SCALE3 ABSPACK scaling algorithm, was applied. The structure was solved by ab initio procedures of the SIR 2014 program [69]. The asymmetric unit, in the monoclinic space group P2_1_, is composed of one *bis*-Fc-dipeptide and one co-crystallized CDCl_3_ molecule. Refinement was carried out by full-matrix least-squares procedures on F^2^, using all data, by application of the SHELXL-2014 program [70], with anisotropic displacement parameters for all of the non-H atoms. The cyclopentadienyl rings of the two Fc moieties were constrained to the idealized geometry. The co-crystallized CDCl_3_ molecule is highly disordered. It was eventually modeled with the three Cl atoms at two sets of positions with population parameters of 0.55 and 0.45, respectively. Restraints were applied to the bond distances and bond angles involving the Cl atoms, as well as to the anisotropic displacement parameters of the latter atoms and of the carbon atoms of the Fc moieties. H-Atoms were calculated at idealized positions and refined using a riding model. CCDC 2193934 contains the supplementary crystallographic data for **2**. The data can be obtained free of charge from The Cambridge Crystallographic Data Centre via www.ccdc.cam.ac.uk/structures (accessed on 14 September 2022).

### 3.3. Cyclic Voltammetry

The experiments were performed in an air-tight three-electrode cell connected to a vacuum/argon Schlenk line. Dichloromethane solvent was pre-dried with anhydrous calcium chloride, refluxed over calcium hydride and distilled under a stream of Argon. Solvent and *n*Bu_4_NPF_6_ were degassed in a Schlenk flask by manifold freeze–pump–thaw cycles and transferred by cannula in the cell. The reference electrode was a SCE (Tacussel ECS C10) separated from the solution by a bridge compartment filled with the same solvent/supporting electrolyte solution used in the cell. The counter electrode was a platinum spiral with around 1 cm^2^ apparent surface area. The working electrode was a disk obtained from the cross section of a gold wire with 0.5 and 0.125 mm diameter sealed in glass. Between successive scans, the working electrode was polished on alumina according to standard procedures and sonicated before use. An EG&G PAR-175 signal generator was used. The currents and potentials were recorded on a Lecroy 9310 L oscilloscope. The potentiostat was home-built with a positive feedback loop for compensation of the ohmic drop [71].

### 3.4. UV–Vis Analysis

Solution absorption spectra in the UV–vis region were recorded at 293 K with a JASCO V770 double-beam spectrophotometer using quartz cells with 1 mm optical path. A spectrum of CH_2_Cl_2_ (baseline) was recorded under the same conditions. Oxidation was achieved in an air-tight container connected to a vacuum/argon line by incremental addition of oxidizing agent solution (ferrocenium[BF_4_]/CH_2_Cl_2_ from 0.1 to 1.0 equivalents and acetylferrocenium[BF_4_]/CH_2_Cl_2_) from 1.2 to 2.0 equivalents. HPLC-grade CH_2_Cl_2_ ≥ 99% was purchased from Carlo Erba and distilled from CaH_2_ (≥97% powder Sigma-Aldrich). Cells with path lengths of 1 mm (with quartz windows) were used.

### 3.5. FT–IR Analysis

Solution FT–IR absorption spectra were recorded at 293 K using an FT-IR Nicolet Nexus 670 spectrophotometer, nitrogen flushed, equipped with a sample shuttle device, at 2 cm^−1^ nominal resolution, averaging 25 scans. Solvent (baseline) spectra were recorded under the same conditions; the solvent used was spectrograde CHCl_3_-d_1_ (99.8%, d). HPLC-grade CH_2_Cl_2_ ≥ 99% was purchased from Carlo Erba and distilled from CaH_2_ (≥97% powder Sigma-Aldrich). For spectral elaboration, the software SpectraCalc provided by Galactic (Salem, MA, USA) was employed. Cells with path lengths of 1 mm (with CaF_2_ windows) were used.

### 3.6. Circular Dichroism

The CD spectra were obtained on a Jasco (Tokyo, Japan) model J-715 spectropolarimeter. Cylindrical fused quartz cells (Hellma, Müllheim, Germany) of 1 cm path length were used. The values are expressed in terms of Δ*ε* (L mol^–1^ cm^–1^). Spectrograde CDCl_3_ (99.8% D, Fluka) was used as a solvent. 

## 4. Conclusions

The comparison among three bis-ferrocenyl, end-capped peptide series, namely Fc-CO–(Aib)_n_–NH–Fc (*n* = 1–5) [25,26], Fc–CO–(ΔAla)_n_–NH–Fc (*n* = 1–4) [28] and Fc–CO–[L−Dap(Boc)]_n_–NH–Fc (**1**–**4**) here reported, highlights the important role that peptide helices play in charge transmission, and the influence of the peptide skeleton on the redox properties of these ferrocenyl-conjugated foldamers. Indeed, when the Fc–peptide–Fc series are based on Aib or on side-chain-protected Dap residues, the conformational analysis in the solid state (X-ray crystal structures) and in solution (IR and 2D NMR) strongly indicates that these peptides adopt a 3_10_-helical structure. In this peptide conformation, the orientation of the carbonyl groups produces a macrodipole with the positive pole on the N-terminal amino acid and the negative pole on the C-terminal amino acid. In contrast, the ΔAla series adopts the fully extended conformation (2.0_5_-helix), possessing no dipole moment.

As a consequence, the cyclic voltammetry experiments on the Aib and _L_-Dap(Boc) homopeptides showed a positive or negative influence on the redox potential of the two terminal Fcs, depending on the peptide end at which they are attached: by increasing the peptide length, NH−Fc is oxidized more easily, whereas the oxidation of Fc–CO becomes more difficult. On the contrary, in the ΔAla series, the terminal Fcs show no or slight variation in their *E*_1/2_.

The modifications of the IR bands upon chemical oxidation also supports this conclusion. Indeed, the transfer of positive charge along the peptides occurs through the H-bond network of a 3_10_-helical structure, even in the longest conjugate, **4^+^**. In addition, the second charge does not affect the efficiency of the **_L_**-Dap(Boc)-based 3_10_-helices in transmitting the positive charge along the peptide chain. Remarkably, the **_L_**-Dap(Boc) peptides show an efficacy even greater than that of Aib peptides [26] in conveying the positive charge. This effect is noteworthy as Aib is a strongly helicogenic amino acid, known to form homopeptides that are very efficient at transferring charge and/or electrons.

Finally, CD experiments on **1**–**4** peptides evidenced “exciton-coupled” CD among the two Fc groups, depending not only on the Fc–Fc distance, but also on the reciprocal orientation of the two aromatic units, and, crucially, on the peptide H-bond scheme.

## Data Availability

CCDC 2193934 contains the supplementary crystallographic data for this paper. The data can be obtained free of charge from The Cambridge Crystallographic Data Centre via www.ccdc.cam.ac.uk/structures (accessed on 14 September 2022).

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
