# Peer review of "Helical versus Flat Bis-Ferrocenyl End-Capped Peptides: The Influence of the Molecular Skeleton on Redox Properties"

_molecules, 2022, doi:10.3390/molecules27186128_

Round 1
Reviewer 1 Report
In this manuscript, the authors describe the correlation between conformation and electrochemical properties of bis-ferrocene peptides. I recommend this paper for publication in Molecules.
Issues:
Line 41, example of bis-ferrocenyl end-capped peptides M. Nuskol et al., Polyhedron 161 (2019) 137–144 should be cited
N- and C-terminus should be written in italic, as well as t in t-butyl…..
Stereochemical descriptors (L-, D-) should be written in a smaller font
Quality of Fig.1.?
I can’t find the yields of target compounds
Line 125, 1 in 1H should be written in superscript
To make the paper easier to read, numeration should be presented on the structure of at least one compound
Since ferrocene peptides with two organometallic chromophores are poorly described in the literature, I think it would be very interesting to record their CD spectra. Also, to see the influence of growing peptide chain (“degree” of folding) on CD-signal.
Author Response
We would like to thank the reviewer that has contributed to improve the quality of the manuscript with their comments.
We addressed the minor points raised and provided a point-by-point response in the attached file.
Best regards

Reviewer 2 Report
This paper by Santi et al. describes the effect of the number of end-capped DAP(Boc) peptides on the structural and electronic effects along the corresponding peptide skeleton, probed by two terminal ferrocene groups which oxidize at potential values sensitive to the average dipolar moment induced by the orientation of the peptide carbonyl groups in helical conformation. These results are compared to previous studies carried out in the same group on other peptides (e.g more rigid D Ala peptides) either on mono- or bis-ferrocenyl functionnalized peptides. The structural XRD and NOESY study is convincing. This work is of quality and proposes an interesting focus on the conformation / electronic properties relationship. In my view, it is suitable for publication in Molecules, providing some minor corrections are made :
1/ In Figure 6, the cyclic voltammograms are claimed to be normalized vs. concentration. So I don't understand the iv-1C-1 equation formula in the caption. It may be confusing if what is actually represented along the Y axis is the current/concentration ratio (it looks like the current is also normalized with respect to the scan rate - V-1. In that case, the reliable parameter would be V-1/2 to highlight a diffusion-controlled current variation).
2/ Figure 6 and comment in lines 195-200 : the peak current decrease between n=1 and n=4 is too high to be only ascribed to the difference in diffusion coefficients, supposedly following a Stokes Einstein evolution with 1/r (r = radius of the particle). In analog studies, chain length a priori does not affect current peaks that much (see e.g an article on viologen redox active polymers at 10.1021/acs.chemmater.6b02825 ). Therefore, I wonder whether 1) the concentration of the peptides in Figure 6 is known with sufficient accuracy to make the current/concentration normalization reliable and 2) the comment on the Stokes Einstein dependence should be moderated, namely because the equivalent radius r should not linearly vary with peptide length if an helical conformation is claimed. Maybe the authors could bring some precisions that support this assertion or moderate this comment.
Author Response

(The authors gave the same response as above.)
